# Zambian Peer Educators for HIV Self-Testing (ZEST) study: rationale and design of a cluster randomised trial of HIV self-testing among female sex workers in Zambia

Catherine E Oldenburg,[1,2] Katrina F Ortblad,[3] Michael M Chanda,[4] Kalasa Mwanda,[4] Wendy Nicodemus,[4] Rebecca Sikaundi,[4] Andrew Fullem,[5] Leah G Barresi,[1] Guy Harling,[3,6] Till Bärnighausen[3,7,8]

For numbered affiliations see end of article.

**Correspondence to**
Dr Catherine E Oldenburg; catherine.oldenburg@ucsf.edu

## ABSTRACT

**Background:** HIV testing and knowledge of status are starting points for HIV treatment and prevention interventions. Among female sex workers (FSWs), HIV testing and status knowledge remain far from universal. HIV self-testing (HIVST) is an alternative to existing testing services for FSWs, but little evidence exists how it can be effectively and safely implemented. Here, we describe the rationale and design of a cluster randomised trial designed to inform implementation and scale-up of HIVST programmes for FSWs in Zambia.

**Methods:** The Zambian Peer Educators for HIV Self-Testing (ZEST) study is a 3-arm cluster randomised trial taking place in 3 towns in Zambia. Participants (N=900) are eligible if they are women who have exchanged sex for money or goods in the previous 1 month, are HIV negative or status unknown, have not tested for HIV in the previous 3 months, and are at least 18 years old. Participants are recruited by peer educators working in their communities. Participants are randomised to 1 of 3 arms: (1) direct distribution (in which they receive an HIVST from the peer educator directly); (2) fixed distribution (in which they receive a coupon with which to collect the HIVST from a drug store or health post) or (3) standard of care (referral to existing HIV testing services only, without any offer of HIVST). Participants are followed at 1 and 4 months following distribution of the first HIVST. The primary end point is HIV testing in the past month measured at the 1-month and 4-month visits.

**Ethics and dissemination:** This study was approved by the Institutional Review Boards at the Harvard T.H. Chan School of Public Health in Boston, USA and ERES Converge in Lusaka, Zambia. The findings of this trial will be presented at local, regional and international meetings and submitted to peer-reviewed journals for publication.

**Trial registration number:** Pre-results; NCT02827240.

### Strengths and limitations of this study

▶ This study will be one of the first reporting the effectiveness and safety of an HIV self-testing (HIVST) intervention among female sex workers (FSWs).
▶ The results will allow for assessment of the feasibility and acceptability of HIV self-test kits in general.
▶ The results will provide evidence on whether HIVST is used by FSWs if it is accessible via existing distribution points within the health system.
▶ Limitations of this study include the self-reported nature of many of the outcomes, which may lead to social desirability biases.
▶ To overcome social desirability biases in the reporting of outcomes, in particular the use of HIVST and HIV status knowledge, we use several innovative measurement approaches.

## INTRODUCTION

Female sex workers (FSWs), defined as women who exchange sex for money or goods, bear a disproportionate burden of the HIV epidemic globally.[1 2] Global HIV prevalence among FSWs has been estimated at nearly 12%, with HIV prevalence rising to more than 30% for FSWs living in countries with medium and high background HIV epidemics.[1] In sub-Saharan Africa in 2011, an estimated 17.8% of all infections in women were attributed to engagement in sex work.[3] Vulnerability to HIV among FSWs occurs at multiple levels,[4] which potentiates the HIV epidemic in this population.[2 5] Determinants of HIV among FSWs occur at the individual level (behavioural and biological factors, such as sexually transmitted infections), dyad

and partner level (such as power dynamics within relationships), work environment level (organisational structures where individuals work, including venue policies and availability of condoms in venues),[6] and community and macrostructural levels (including community structure and organisation and legal and policy environments).[2]

In 2014, the Joint United Nations Programme on HIV and AIDS (UNAIDS) put forth a new global target for controlling and ultimately ending the HIV epidemic.[7] The 90-90-90 target is a three-part target that aims to have 90% of people living with HIV aware of their status, 90% of those diagnosed on treatment and 90% of those on treatment virally suppressed by the year 2020, ultimately resulting in viral suppression for 73% of HIV-infected individuals worldwide. Such a strategy requires regular HIV testing for individuals who are at ongoing risk of HIV, including FSWs. However, significant gaps remain in HIV testing coverage in sub-Saharan Africa.[8] Given the large burden of HIV among FSWs and the unique barriers to HIV prevention faced by this population, evidence of strategies that are specifically designed for this population is urgently needed.

Stigma and discrimination can affect access to HIV testing among FSWs.[9][10] FSWs may be affected by multiple intersecting forms of stigma. Stigma is broadly defined as negative attitudes, relative powerlessness and loss of status related to a particular characteristic.[9] Stigma can be experienced as enacted (explicit actions such as derogatory language, active discrimination or assault), perceived (the expectation of enacted stigma) or self-stigma (internalisation of stigma).[10] For FSWs, stigma can be due to engagement in sex work itself, or from HIV stigma, particularly in contexts of high HIV burden, and can come from family or other community members, partners and healthcare providers. These experiences of stigma may be exacerbated by unequal power dynamics, poverty and financial reliance on sex work, and experiences of violence. Taken together, the multidimensionality of experiences of stigma and discrimination may greatly influence uptake of HIV testing. Interventions that mitigate stigma as a barrier to HIV testing may be especially powerful for FSWs.

Oral HIV self-testing (henceforth, HIVST) is an alternative to clinic or other healthcare provider testing (eg, home-based HIV testing and counselling) that may overcome some barriers to traditional HIV testing strategies for some populations.[11–14] HIVST has been shown to be acceptable in a variety of populations globally,[11] but very little data exist on the acceptability and feasibility of HIVST among FSWs.[13] For FSWs in particular, HIVST may be a particularly good option as it offers privacy protection, efficiency and flexibility. FSWs are a vulnerable population regarding potential harms and threats to livelihood if others are to learn they are HIV positive. HIVST may offer an additional degree of privacy by allowing women to test in a place of their choosing. HIVST may be efficient for FSWs. Current

recommendations indicate that FSWs should ideally test for HIV every 3 months. HIVST could reduce the burden of transport to clinics for traditional testing. Finally, HIVST is flexible. Existing clinics may not offer HIV testing services during hours which are feasible for FSWs, who often sleep during the day and work during the night. HIVST can overcome each of these limitations by allowing a woman to test at a time and a place that is convenient and acceptable to her.

A rapid oral HIVST has been approved in the USA, which uses fluid from the oral mucosa and is read in 20 min.[15] The test, OraQuick (OraSure Technologies, Bethlehem, Pennsylvania, USA), has a sensitivity of 91.7% and a specificity of 99.9% among untrained users.[16] All individuals who test positive with this test are encouraged to get a blood-based confirmatory test. Among men who have sex with men in Seattle, a randomised controlled trial demonstrated a significant increase in HIV testing coverage among individuals receiving HIVST.[17] Although HIVST may be an acceptable alternative to regular HIV testing for FSWs, minimal evidence of the use of HIVST among FSWs exists.[18] The acceptability, uptake, and effectiveness and safety of HIVST among FSWs remain largely unknown.

Here, we describe a cluster randomised controlled trial that addresses this critical gap in the HIVST evidence for FSWs. In the trial, we test two health service delivery approaches, a direct distribution system where an FSW participant is directly given an HIVST by a peer, and a fixed distribution system where the FSW participant collects the HIVST from a fixed point such as a pharmacy or health post. The trial seeks to determine the effectiveness and safety of these two HIVST delivery approaches, compared with standard of care, for improving HIV testing coverage and knowledge of HIV status.

## METHODS/DESIGN
### Theory of change
The intervention tested in the Zambian Peer Educators for HIV Self-Testing (ZEST) study, as well as data collected throughout the course of the study, was guided a priori by a theory of change developed through mental models and deductive development.[19] Briefly, we theorised that the distribution of HIV self-test kits via peer educators would lead to improved status knowledge by reducing barriers to HIV testing such as stigma hours of clinic operation. Enacted or perceived sex work stigma from healthcare providers and from the community may be addressed by HIVST, by allowing individuals to test for HIV in private without fear of being seen in the clinic and without fear of judgement from providers. This would lead to improved uptake of HIV testing, which would lead to improved knowledge of status and ultimately reduce time to linkage to care. However, it is also possible that a community-based intervention such as HIVST could be unsuccessful if individuals are concerned about others discovering their HIV status.

## Study design

This is a unmasked cluster randomised trial. Individuals are enrolled as members of a peer educator group, with a target enrolment of six FSWs per peer educator (with five allowed if there are recruitment difficulties). The peer educator–FSW group is randomised in a 1:1:1 fashion to one of the three study arms: (1) peer educator distribution of HIVST directly to participants (direct distribution), (2) peer educator distribution of a coupon valid for an HIVST at a fixed distribution point (fixed distribution) or (3) standard of care. Participants will be followed for 4 months from enrolment, with follow-up assessments at months 1 and 4 (figure 1).

## Specific aims

The specific aims of this trial are to: (1) establish the effectiveness of two different distribution approaches (direct and fixed distribution) for increasing HIV testing coverage among FSWs; (2) establish the effectiveness of the two distribution approaches on increasing HIV status knowledge among FSWs; (3) measure HIVST uptake by distribution approach and (4) measure linkage to care by the different approaches. Additionally, the trial will measure the impact of HIVST using the two distribution approaches on a number of important safety end points. The trial design will allow for a comparison of the two distribution approaches to each other as well as to standard HIV testing.

## Study oversight

An independent Scientific Oversight Committee (SOC) oversees the data collected as part of this study. The SOC contains members with expertise in HIV epidemiology, statistics, ethics and female sex work. The SOC reviews any reports of serious adverse events during the course of the trial, and will convene a meeting after approximately one-third of the participants have reached their 1-month visit. Owing to the short duration of the study, there are no formal statistical stopping rules.

## Setting

This study is taking place in three border and transit towns in Zambia: Chirundu, Livingstone and Kapiri (figure 2). Chirundu and Livingstone are located on the Zambia–Zimbabwe border, and are major transportation points for people and goods. Kapiri is north of the capital, Lusaka, and is a transit hub, with a large weigh station where many truckers stop for the night or longer. Study headquarters and coordination for the three sites is located in the capital, Lusaka.

## Recruitment

Participants are recruited by peer educators. The peer educators are current or former FSWs themselves who were recruited by study staff and who have undergone a 2-day training in study procedures and HIVST. Peer educators, who are not considered as participants in this study, were recruited from current and former sex work organisations working in each of the study towns. Interested potential participants are referred to study staff after the peer educator describes the study to them. A research assistant then conducts a phone screening to determine whether or not the participant is eligible. Formal eligibility assessments are done in person at

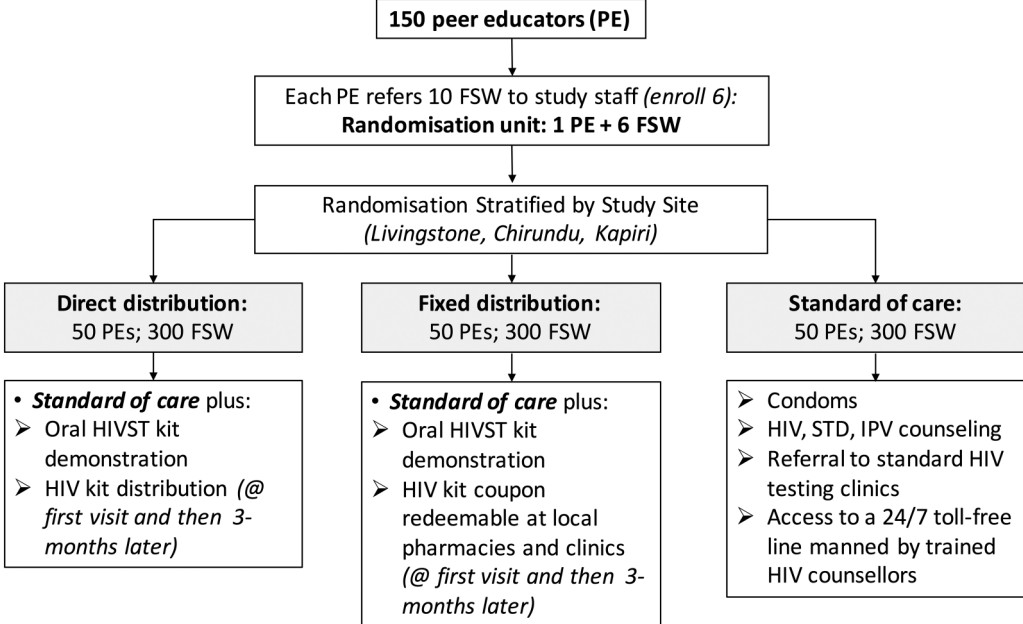

**Figure 1** Flow diagram of study enrolment, randomisation and intervention arms. FSW, female sex worker; HIVST, HIV self-testing; STD, sexually transmitted disease; IPV, intimate partner violence.

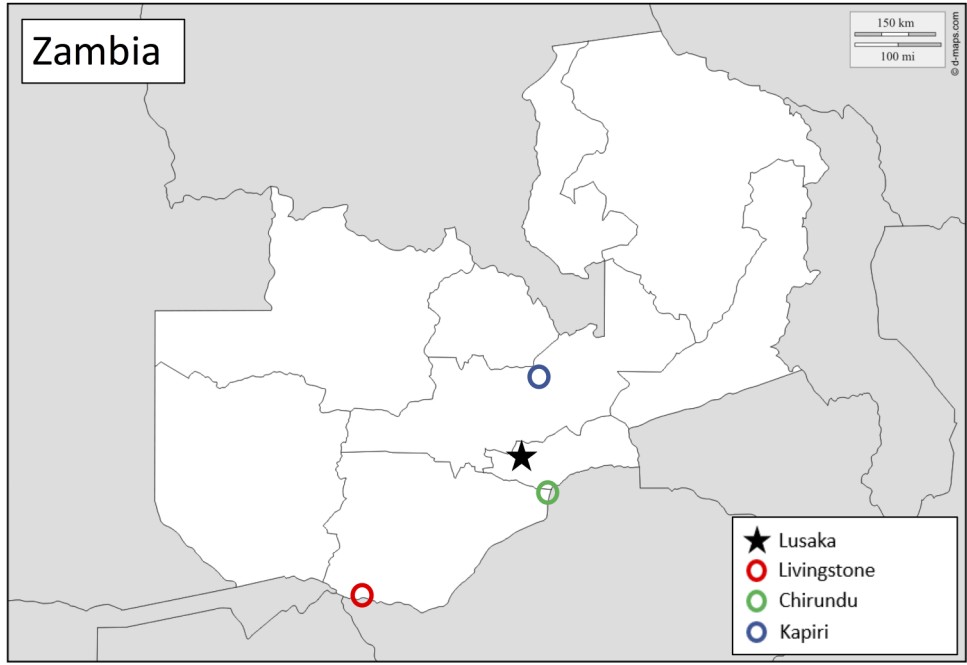

**Figure 2** Map of study sites in Zambia. The black star indicates Lusaka, the site of study coordination and headquarters. The circles indicate locations of study sites, including Kapiri (blue), Chirundu (green) and Livingstone (red).

which point the research assistant and eligible individuals complete an informed consent procedure. Written informed consent is obtained from all participants.

### Inclusion and exclusion criteria

Complete inclusion and exclusion criteria are summarised in table 1. Participants must be women 18 years of age or older who self-report exchanging sex (including vaginal, anal and/or oral) for money or goods in the past month.

Individuals who are currently residing in the PopART study catchment area[20] in Livingstone will be excluded from this study, in addition to any individual who reports concomitant enrolment in another HIV

prevention study. The PopART study includes a household HIV testing programme, which would bias outcomes in this study if individuals were being regularly tested as part of another HIV prevention study.

### Randomisation

Peer educator groups (the 5–6 women recruited by a particular peer educator and enrolled in the study) are randomised as a unit. Group randomisation occurs after all members of the group have been enrolled in the study and completed their baseline questionnaire. Groups are randomised to one of the three study arms using a computer-generated randomisation list in random blocks of 3, 6 and 9 and stratified by study site

| Table 1 Study inclusion/exclusion criteria | |
| --- | --- |
| **Inclusion criteria** | **Exclusion criteria** |
| ▶ 18 years or older at enrolment<br>▶ Reports exchanging sex (vaginal, oral and/or anal) for money or goods at least once in the past month<br>▶ Self-reported HIV negative and no recent HIV test (<3 months) OR HIV status unknown<br><br>▶ Permanent residence in the study town of enrolment (Livingstone, Chirundu, Kapiri)<br><br><br>▶ Willing to participate in peer education sessions and study assessments over a 4-month study period | ▶ <18 years at enrolment<br>▶ Has not exchanged any form of sex in the past month<br>▶ Self-reported to be living with HIV<br>▶ Self-reported HIV negative but tested within the past 3 months<br>▶ Planning to move out of the geographical area within 4 months<br>▶ Living in the PopART catchment area (Livingstone only)<br>▶ Meets inclusion criteria but does not wish to participate<br>▶ Concurrently participating in another HIV prevention study |

**Table 2** Outline of study visits

| Study time point | Visit by | Activities |
|---|---|---|
| Recruitment | Peer educator | ▸ Peer educator discusses study with potentially eligible participant and refers them to study staff |
| Enrolment | Research assistant | ▸ Assessment of eligibility<br>▸ Informed consent (if eligible)<br>▸ Baseline quantitative survey<br>▸ Baseline qualitative survey (~5%) |
| Randomisation | Research assistant | ▸ Randomisation of peer educator and their cohort of participants once all group members are enrolled |
| Intervention visit 1 (week 0) | Peer educator | ▸ Group-based intervention<br>▸ HIV prevention counselling and distribution of condoms<br>▸ Training on HIV self-test use (in HIVST arms)<br>▸ Distribution of HIV self-tests or coupons (in HIVST arms) |
| Intervention visit 2 (weeks 2–3) | Peer educator | ▸ Screening for adverse events<br>▸ Discussion of any difficulty with HIV self-test use (in HIVST arms)<br>▸ Referral to care and standard HIV testing<br>▸ Distribution of condoms |
| One-month visit (weeks 4–5) | Research assistant | ▸ One-month quantitative survey<br>▸ One-month qualitative survey (~5%)<br>▸ Screening for adverse events |
| Intervention visit 3 (weeks 6–7) | Peer educator | ▸ Screening for adverse events<br>▸ Discussion of any difficulty with HIV self-test use (in HIVST arms)<br>▸ Referral to care and standard HIV testing<br>▸ Distribution of condoms |
| Intervention visit 4 (weeks 10–12) | Peer educator | ▸ Screening for adverse events<br>▸ Discussion of any difficulty with HIV self-test use (in HIVST arms)<br>▸ Referral to care and standard HIV testing<br>▸ Distribution of condoms<br>▸ Distribution of second HIV self-test or coupon (in HIVST arms) |
| Four-month visit (weeks 16–18) | Research assistant | ▸ Four-month quantitative survey<br>▸ Four-month qualitative survey (~5%)<br>▸ Screening for adverse events<br>▸ HIV status assessment |

HIVST, HIV self-testing.

(Livingstone, Chirundu and Kapiri). Randomised study arm assignments are placed in an opaque envelope, and once all participants in a peer educator group have been enrolled, the research assistant obtains the envelope from the research coordinator or the site coordinator and opens the envelope with the peer educator.

### Study procedures

An overview of the study procedures and activities is summarised in table 2 and figure 3.

### Peer educator visits

In all study arms, peer educators will meet with participants at least four times (approximately monthly) over the course of the study. The first peer educator intervention visit (intervention visit 1) will be a group-based visit, and includes simple HIV prevention information, information on where to go for HIV testing and condom provision. The initial group-based intervention is organised by the peer educator with assistance from the research assistant, and occurs in a private location of mutual convenience for the group members (such as a study office, or near a work place). In the HIVST arms, peer educators will provide a brief overview of how to use the self-test kits, whom to contact if participants have problems or concerns, and the importance of as well as how to obtain additional care, including confirmatory testing and linkage to care. Follow-up visits (intervention visits 2–4) with peer educators comprise brief one-on-one check-in visits, during which peer educators will distribute condoms, discuss any issues participants have with the HIVST kits or other elements of the study, and screen for adverse events including intimate partner violence. Between visits, peer educators will be available should participants need help with any issue relating to safe sex, HIV, violence or any other elements of the study. Peer educators do not specifically discuss HIV status with any participant unless the participant seeks out the peer educator for advice. Participants are given alternatives should they need help with HIVST or another element of the study but do not want to disclose their HIV status to the peer educator. In particular,

**Figure 3** SPIRIT flow chart of study assessments. HIVST, HIV self-testing.

| | STUDY PERIOD | | | | | | |
| --- | --- | --- | --- | --- | --- | --- | --- |
| | Enrolment | Allocation | | Post-Allocation | | | Close-Out |
| **TIMEPOINT** | | 0 | 2 weeks | 1 month | 6 weeks | 10 weeks | 4 months |
| **ENROLMENT:** | | | | | | | |
| **Eligibility screen** | X | | | | | | |
| **Informed consent** | X | | | | | | |
| **Allocation** | | X | | | | | |
| **INTERVENTIONS:** | | | | | | | |
| *HIVST Kit/Coupon Distribution* | | X | | | | X | |
| *Peer Educator Visit* | | X | X | | X | X | |
| **ASSESSMENTS:** | | | | | | | |
| *Demographics, HIV testing history, sexual behaviors, reproductive health, HIV knowledge, empowerment* | X | | | | | | |
| *HIV testing history, sexual behaviours, HIV knowledge, empowerment* | | | | X | | | X |
| *HIV status knowledge* | | | | | | | X |
| *HIV self-test kit use* | | | | | | | X |
| *Adverse Events* | | | X | X | X | X | X |

participants in all study arms also have access to an anonymous 24-hour hotline provided via the study. Participants will be instructed by the research assistant that they can call the 24-hour hotline at any time if they have difficulty with using the HIV self-test kit, need counselling related to HIV testing or gender-based violence, or need any other form of assistance over the duration of the study.

### Direct distribution arm
In the direct distribution arm, peer educators will directly distribute HIVST kits to participants during the first intervention visit. The HIVST kit consists of a single OraQuick in-home HIV self-test and instructions translated into English, Nyanja and Bemba. The instructions are a step-by-step pictorial and written guide for using the test and interpreting the results.

Peer educators offer a second HIVST kit to participants in the direct distribution arm at the fourth intervention visit. There is no HIV status requirement for distribution of a second test. The second test is distributed at the last peer educator visit, 3 months after the first, to minimise contamination of study arms.

### Fixed distribution arm
In the fixed distribution arm, peer educators give participants a coupon which they can exchange for an HIVST kit at a participating fixed distribution facility. These facilities include drug stores and health posts that have agreed to participate in the study and have received basic training on HIVST. Staff at the fixed distribution points collect the coupons when they are exchanged for an HIVST kit and give them to research staff on a weekly basis. These coupons include a bar code that is scanned by study staff to record which participants collected the HIVST kit.

Peer educators offer a second coupon to participants at the fourth intervention visit to individuals, similar to the direct distribution arm, which can be used to obtain the second HIVST kit.

### Standard of care arm
In the standard of care arm, peer educators give participants information about existing HIV testing services where participants can go for testing. In Livingstone, Chirundu and Kapiri, available options generally include government facilities, non-profit faith-based facilities and private facilities. While clinics designed specifically for FSWs were previously in operation in these cities, funding constraints forced these facilities to close in 2015. The information that participants receive about HIV testing facilities is customised to the city of their recruitment, and identical information is given to participants in each of the three arms.

In all study arms, participants will receive a linkage to care card from their peer educator that they will be requested to bring if they seek confirmatory testing and HIV care.

**Table 3** Summary of study end points

| End point | Operationalisation |
|---|---|
| *Primary effectiveness end point* | |
| Tested for HIV in the past month | Recent HIV testing measured by asking participants when they last tested and where (in all arms of the study) |
| *Secondary effectiveness end points* | |
| Use of HIV self-test | Measured by buying back unused HIV self-tests at the 4-month visit |
| Awareness of HIV status | Measured using a three-step approach: (1) asking participants if they know what their status is; (2) offering participants a small financial gift if they can correctly tell the interviewer what their HIV status is; 3) confirming HIV status with a rapid test |
| Linkage to care and confirmatory testing | Measured in two ways: (1) during follow-up visits, asking participants if they received confirmatory testing and linked to care; and (2) collection of referral cards linking individual HIV self-tests to individuals via a unique identification number |
| Sexual behaviours | Measured via CAPI, including number of sexual behaviours, event-level sexual behaviour data, and condom use with primary and casual partners |
| *Safety end points* | |
| Misuse of HIV self-tests | ▶ Including difficulty conducting the test (ie, mistakes in taking the test, incorrect use of components of the test), difficulty reading the test<br>▶ Identified through interview and ongoing consultation with peer educators |
| Intimate partner violence | ▶ Measured through surveillance and interviews by research assistants<br>▶ Any intimate partner violence (including verbal, physical or sexual) will be documented and reported |

CAPI, computer-assisted personal interview.

## Data collection

### Baseline survey: enrolment

At the enrolment visit, all participants complete a quantitative questionnaire administered by a research assistant. Quantitative data are collected directly on a tablet. All assessments are conducted in a private room at a time and location that is convenient to the participant. The baseline survey covers broad topics including demographics and household composition, sex work and professional history, reproductive health and contraception, sexual behaviours, HIV testing history, attitudes towards HIVST, access to healthcare, and measures of substance use,[21] depression,[22] social support, self-efficacy,[23] and empowerment.[24]

### One-month and 4-month visits

Follow-up surveys will be conducted at the 1-month and 4-month visits from intervention visit 1 (see table 2). As with the baseline survey, all quantitative data will be collected via a tablet, and qualitative data (4-month visit only) are collected via an audio recorder. Participants in all arms of the study will be asked about their recent HIV testing history, sexual behaviours, access to healthcare and linkage to care. In the HIVST arms of the study, participants will additionally be asked about their experiences with the HIVST kit, including if they used it and details about their experience using the test, as well as their experience with the peer educator. Participants in all arms are asked again at 1 and 4 months about HIVST acceptability and perceived likelihood of using such tests in the future.

### Qualitative interviews

Approximately 5% of the sample (15 participants per arm, 45 in total) are randomly selected at baseline to participate in individual qualitative in-depth interviews. During baseline screening, a random subset of participants will be asked if they would be interested in participating in the qualitative phase of the study. Individual in-depth interviews are conducted by trained research assistants at baseline and at the 4-month follow-up visit (table 2). All qualitative data are collected via an audio recorder. Baseline interviews explore community norms related to HIV testing, multilevel (including individual, structural and interpersonal) barriers to accessing HIV testing and other healthcare, experiences in sex work, and baseline perceptions of HIVST. At follow-up, qualitative interviews focus on experiences using the HIV self-test for individuals who are in one of the two intervention arms, and individuals in all arms will be asked about their perceptions of the HIV self-test and how HIVST might be implemented at scale. In all arms at follow-up, the interviews will explore empowerment and self-efficacy related to HIV testing and HIV status disclosure.

## End point measurement

Study effectiveness and safety end points are summarised in table 3. The primary effectiveness end point for the study is the proportion of participants who tested in the previous month as assessed at the 1-month follow-up visit. Secondary effectiveness end points include testing at 4 months, awareness of HIV status, HIVST kit use (restricted to the HIVST arms), linkage to care and adverse events. Safety end points include intimate partner violence and mental health.

### Recent HIV testing

Recent HIV testing is assessed by self-report at the 1-month and 4-month visits. Participants are asked when

their last HIV test was, and then asked where their test was to determine if their last test was an HIVST or other.

### HIV status knowledge

Correct reporting of HIV status via a self-report is subject to social desirability bias. To accurately measure whether an individual knows their HIV status, they must both correctly report knowledge of their HIV status and be willing to take an HIV test. To overcome some of the bias in ascertaining self-reported HIV status, we use a three-step approach. First, participants are asked if they know their current HIV status. They are then asked to tell the research assistant their current HIV status, and are told to take their best guess if they do not know. Participants are then asked to take a rapid HIV test, and are counselled that this test is entirely voluntary. To reduce bias in self-reporting HIV status, participants are told that they will receive a small gift (worth ~US$1) if they correctly report their status. However, after the assessment, all participants will receive the gift regardless of their reported status; thus, the cash transfer is perceived as conditional but is actually unconditional for ethical reasons. Pretest and post-test counselling will be available to all individuals who participate in the HIV status assessment. This assessment is performed only at 4 months.

### HIVST kit use

Actual use of the HIVST kit may also be subject to social desirability bias. Participants may want to report that they used the HIVST kit when they did not for many reasons, including a desire to please the researchers, a sense of obligation or misperception about what it means to use the test.[25] As an attempt to reduce bias in ascertainment of actual use of the HIVST kits, research assistants will ask participants to buy back any unused HIVST kits at the end of the study with the justification that the study has now reached completion. Participants will be offered ~US$1 in exchange for any unused test kits in their original packaging. Prior to this offer, participants are not told that researchers will buy back tests at the end of the study (this information is also not included in the informed consent). To avoid gossip among study participants, the first buy-back will not happen until at least 1 month after the last test kit has been distributed (with the assumption that all participants would have used the test kit at that point if they were planning to).

### Stigma and empowerment

Stigma and empowerment are measured both quantitatively and qualitatively at baseline and 4 months. Empowerment is measured additionally at 1 month. Quantitatively, sex work stigma is measured via a six-item scale that covers enacted, perceived and self-stigma,[26] and HIV stigma is measured by a nine-item scale measuring attitudes towards individuals living with HIV.[27] Qualitatively, stigma measurements include questions related to barriers to accessing care, interactions with police and experiences working as a sex worker. We hypothesise that at baseline individuals who report a greater amount of stigma will have reduced HIV testing, and that these individuals will have greater uptake of HIVST. Empowerment is defined as the process by which disenfranchised individuals acquire the ability to make choices.[24] Quantitatively, empowerment is measured with a five-item scale addressing sex worker agency and power within.[24] Qualitatively, empowerment is measured as the ability to get things the participant needs, to change her situation and control over elements such as condom use. We hypothesise that access to HIVST will lead to an improved sense of empowerment.

### Adverse events

Adverse events are monitored and documented throughout the study. The primary adverse events of concern are psychological harm as a result of use of the HIV self-test (eg, if participants test positive for HIV and are not prepared for the results). Although UNAIDS has recently indicated no evidence of serious adverse events related to HIVST (ie, self-harm, violence or human rights violations),[28] occurrence of any such events are reported to all principal investigators and the SOC within 24 hours of study staff becoming aware of them. Other adverse events include coercive testing or unintentional or unauthorised disclosure of HIV status, which are similarly reported within 24 hours.

### Statistical analysis

The primary analysis will be intention-to-treat. The primary analysis for the quantitative data will involve a mixed-effects multilevel logistic regression model to account for clustering by peer educator group and study site (participants nested in peer educators nested in study sites). These models will include an indicator term for study arm (fixed distribution, direct distribution or control). Outcomes, including past month HIV testing and correct knowledge of status at 4 months, will be modelled as dichotomous variables.

### Qualitative data analysis

Qualitative data will be analysed using an iterative descriptive approach to characterise and describe the data at their natural level. After all interviews are transcribed and translated, a codebook will be finalised and transcripts will be coded in Dedoose, a cloud-based platform for mixed-methods research.

### Sample size considerations

The sample size calculation was based on the primary end point, testing for HIV in the past month assessed at the 1-month visit, and the secondary end point, proportion of participants who correctly know their status at the 4-month visit. Power calculations were performed using methods for cluster randomised trials, with clustering by peer educator identity. We assume that 50% of

participants will have tested in the past month in the control arm based on previous estimates of HIV testing behaviour among FSWs in Livingstone and Chirundu.[29] [30] We estimate that in the control group, ~70% of participants will correctly identify their HIV status, assuming that the majority of those whose status is unknown will report that they are negative, but that approximately half of these individuals will actually be positive. Assuming a 20% loss to follow-up and with 50 peer educators (clusters) per arm, a sample size of 300 participants per arm (900 total) will yield 89% power to detect a risk ratio of 1.30 for recent testing and 94% power to detect a risk ratio of 1.20 for correct status knowledge, assuming a type I error probability of 0.05 and an intracluster correlation of 0.03.

## Confidentiality

To protect participants' confidentiality, no identifying information will be collected during any quantitative or qualitative assessment and all collected data are de-identified. A unique study identification number will be used to link data from an individual participant. Participant identity codes are kept completely separately from electronic data, in a locked cabinet in study offices. The code will be destroyed on completion of the study.

## Data management

Quantitative data are collected directly on a study tablet via CommCare (Dimagi, Cambridge, Massachusetts, USA), a Health Insurance Portability and Accountability Act (HIPAA)-compliant, cloud-based platform for electronic data collection. Qualitative data are collected via an audio recorder, which are transcribed and translated verbatim. Qualitative audio recordings are destroyed on completion of transcription and translation.

## Dissemination plan

The results of this study will be disseminated to community, policy and scientific stakeholders. A results community meeting will be planned on completion of the study at each study site, during which the results of the study will be announced. We will also hold a meeting with the Ministry of Health in Lusaka to disseminate our findings, and submit a report detailing the findings of the study to inform policy. Finally, results will be shared with the scientific community via presentation at international conferences and journal publication.

## DISCUSSION

Despite the high risk and large burden of HIV among FSWs, HIV testing and HIV status knowledge in this population remain far from universal. A number of complex barriers at multiple levels can limit access to HIV testing for FSWs, including fear of stigma and abuse, fear of loss of livelihood, and the financial and time costs of using traditional HIV testing services. HIVST is a promising alternative HIV testing option for

FSWs, because it allows for a high degree of privacy, efficiency and flexibility in HIV testing. The overarching goal of the ZEST trial is to provide robust evidence for the development of policy for implementation and scale-up of HIVST in Zambia with a specific focus on ensuring that this technology is accessible to FSWs. Working in concert with the Ministry of Health in Zambia and the WHO, this trial was specifically designed to fill a crucial gap in the evidence by answering two questions: (1) Is HIVST an acceptable alternative to standard HIV testing for FSWs? and (2) Will FSWs access HIVST?

Question 1 is addressed through the direct distribution arm. In this arm, we will be able to assess whether individuals directly given a test actually use the test. By directly giving individuals the test, we are assessing best-case scenario uptake, where there are as few barriers as possible for individuals to access the test kit. The uptake results from this arm of the study will give robust evidence on whether HIVST is an acceptable alternative to standard and currently available HIV testing services. Coupled with qualitative data, we will be able to measure both the uptake of HIVST and participant attitudes towards HIVST. This will lead to an in-depth understanding of why participants did or did not use the test, and how participants view the role of HIVST for HIV testing in the future.

Although uptake and acceptability of HIVST is an important piece of evidence for HIVST policy, the design of this study allows for measurement of a more realistic scenario, in which participants must go to a physical distribution point to collect their kit. Question 2 is thus addressed through the fixed distribution arm. This arm will measure the degree to which barriers to acquiring the HIVST kit in the store affect acquisition and use of the test. This will be tested by implementing the project within existing drug stores and health posts, without changing the hours or geographical locations of existing services. The direct and fixed distribution arms will therefore give two complementary pieces of information—first, are participants using the test kits? and second, are they able to access them independently?

A major strength of this study is that it does not rely on clinic-based sampling or recruitment. A potential benefit of HIVST over existing HIV testing services is that it may allow individuals who are not currently engaged in care or who have never sought HIV testing because of barriers to accessing care to test for HIV. HIVST may therefore be a gateway to care for individuals who have previously been difficult to engage in care. By working through peer educators, rather than in clinics, we will access a different subset of the population. Peer educator programmes have been used in a variety of settings with different sex worker populations to increase HIV prevention knowledge.[31–36] In many cases, peer educators have unique access to and rapport with FSWs. This study, therefore, is expected to enrol a diverse population of FSWs that is most likely more

generalisable than a clinic-based sample, while simultaneously allowing access to populations that may not currently access care.

A major concern with HIVST is the potential for intimate partner violence, for example, if sexual partners find out about HIV status through the test. All participants will be screened for intimate partner violence at baseline, 1 and 4 months. Intimate partner violence is very common in the FSW community, and those who have experienced intimate partner violence will be eligible to participate, although the risks of participating are explained fully. Furthermore, during follow-up peer education visits, peer educators will screen for intimate partner violence, and they will also be available between visits should any issues come up. Finally, a 24-hour telephone line will be available to all participants for any issues, including intimate partner violence, and referral will be made to counselling if desired. All reports of intimate partner violence will be reviewed to determine if they are related to HIVST or trial participation. In this study, we do not offer HIVST for FSWs to test with their clients or other sexual partners. While it is possible that partner testing could be a benefit for some participants, it could also put some at increased risk of violence. We will collect detailed quantitative and qualitative data about individuals with whom participants used the test, if anyone. The results of this study are expected to yield evidence of the impact of HIVST on safety end points, in particular intimate partner violence.

Another safety concern with HIVST is the potential for depression, anxiety or suicidality as a result of learning one's HIV status. Structured interviews will contain screening questions for depression, and peer educators will perform a brief screening on depression, anxiety and self-harm at each visit. Finally, participants will be instructed to call the hotline should they need to discuss their HIV result with anyone or receive counselling.

In conclusion, the results of this study are expected to inform policy aimed at implementation and scale-up of HIVST for FSWs in Zambia. This study will begin to address critical gaps in the literature describing evidence of HIV testing interventions for FSWs in Southern Africa. We expect the results of this trial to have a policy impact resulting in increased availability of HIVST in Zambia.

**Author affiliations**
[1]Department of Epidemiology, Harvard T.H. Chan School of Public Health, Boston, Massachusetts, USA
[2]Francis I. Proctor Foundation, University of California, San Francisco, California, USA
[3]Department of Global Health and Population, Harvard T.H. Chan School of Public Health, Boston, Massachusetts, USA
[4]John Snow, Inc, Lusaka, Zambia
[5]John Snow, Inc, Boston, Massachusetts, USA
[6]Research Department of Infection and Population Health, University College London, UK
[7]Africa Health Research Institute, Somkhele, South Africa
[8]Institute of Public Health, University of Heidelberg, Heidelberg, Germany

**Acknowledgements** The authors gratefully acknowledge feedback on and conversations related to our study design from Anna Heard, Annette Brown and Nancy Diaz.

**Contributors** CEO, KFO, MMC, KM, WN, RS, AF, LGB, GH and TB conceived and designed the study. CEO, KFO and LGB created tables and figures. CEO, KFO and MMC wrote first draft of the manuscript. CEO, KFO, MMC, KM, WN, RS, AF, LGB, GH and TB read and critically revised the manuscript.

**Funding** The Zambian Peer Educators for HIV Self-Testing Study is funded by the International Initiative for Impact Evaluation (3ie). CEO is supported in part by the National Institute on Drug Abuse T32-DA013911 and the National Institute of Mental Health R25-MH083620. KFO is supported in part by the National Institute of Allergy and Infectious Disease T32-AI007535. TB is supported in part by the National Institute of Child Health and Human Development R01-HD084233 and the National Institute of Allergy and Infectious Diseases R01-AI124389 and R01-AI112339.

**Competing interests** None declared.

**Ethics approval** Ethical approval was obtained for the Zambian Peer Educators for HIV Self-Testing Study from the Harvard T.H. Chan School of Public Health Institutional Review Board in Boston, Massachusetts and the ERES Converge Institutional Review Board in Lusaka, Zambia.

**Provenance and peer review** Not commissioned; externally peer reviewed.

**Data sharing statement** On completion of the trial, de-identified data will be available through the International Initiative for Impact Evaluation (3ie)'s public data access repository on Dataverse.

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
