## [Reviewer comments · BMJ Open]

ARTICLE DETAILS

TITLE (PROVISIONAL)	The Zambian Peer Educators for HIV Self-Testing (ZEST) Study: Rationale and design of a cluster randomized trial of HIV self-testing among female sex workers in Zambia
AUTHORS	Oldenburg, Catherine; Ortblad, Katrina; Chanda, Michael; Mwanda, Kalasa; Nicodemus, Wendy; Sikaundi, Rebecca; Fullem, Andrew; Barresi, Leah; Harling, Guy; Bärnighausen, Till

VERSION 1 - REVIEW

REVIEWER	Kaja-Triin Laisaar, MD, MPH, PhD Institute of Family Medicine and Public Health, University of Tartu, ESTONIA
REVIEW RETURNED	17-Nov-2016

GENERAL COMMENTS	The study will provide important information on options to improve HIV testing frequency and HIV status knowledge in a high HIV risk group of female sex workers in Zambia. However, the protocol still needs some refinement. Please see below major comments, following the manuscript 'Review Checklist' numeration, and a few minor comments. Major comments: 2. Information in the abstract is not correct, as it is said that "...Participants (N=900) are eligible if they are women who have exchanged sex for money or goods in the previous 3 months,...", while elsewhere in the manuscript the period is limited to "last month" or "past month" (on page 9 and in Table 1, respectively). 4.1. Randomization information provided in the "Study design" paragraph (on page 7) and in Figure 1 (on page 32) is controversial - it remains unclear whether there will be 5 or 6 FSW per peer educator. In case the educators are also considered participants in this study, their recruitment, informed consent provision, etc. should be described in more detail. And even in case the educators are not considered participants, I would like to know more about how they will be recruited -- to get an idea of a potential external validity issue [e.g., educators with organized (establishment-based) vs individual sex work experience recruiting FSW with similar background]. 4.2. While describing the study procedures (starting on page 10) the reader would benefit from referral to Table 2 (in addition to already being referred to Table 1 and Figure 3), currently not mentioned anywhere in the text. I would also suggest to use the study visit names provided in Table 2 in the "Study Time Point" column, as otherwise it would not be obvious (while reading the text) what/when is done. This also applies to Figure 1.
---

	4.3. While reading the overview of the "Peer Educator Visits" (on page 10), I would have expected a few more words about the group-based visits -- how (first of all where) will these be organized. 4.4. Somewhere in the "Study Procedures" paragraph I would also expect a few words about the 24/7 help-line -- will it be created for the study purposes only or is it locally available regardless of the study? 4.5. In the "Data collection" paragraph (on page 13), it should be described in a few words whether data in the follow-up assessments will also be "collected directly on a tablet in the field" (as in the baseline survey). This also applies to the in-depth qualitative interviews. 5. Please see my comment 4.1. 7. How will data from qualitative interviews be analysed? 12. Please see my comment 4.1. regarding external validity. Also, as data management (as described in item 19 of the SPIRIT checklist) and confidentiality (as described in item 27 of the SPIRIT checklist) procedures are not described in detail, one might argue that not all the potential limitations of the study have been discussed in the manuscript. 13. The SPIRIT Checklist seems to be filled out in a bit of hurry. While in general the numbers of pages provided might have shifted because of the title page automatically added to the manuscript submission package, for several items not all the applicable page numbers have been provided. Just two examples: item 5c (22-23), item 22 (11, 16, 20, Tables 2 and 3). Minor comments: a. Regarding individual funding, reported by the authors on page 22, it is not clear whether the support was provided only in the past or will continue throughout the planned study; b. Figure 1 (on page 32) is missing a title; c. The references should be reviewed, as:  - in case of more than three authors, in some cases all the authors have been listed, while in others 'et al.' is used (e.g., reference 2 vs reference 3); - in several cases the number of the journal issue is missing (e.g., reference 8); - in cases the reference is available online, a link might be provided (e.g., reference 11). d. There seem to be a few typos in the text:  - the sentence on page 16 on lines 24-25 should, perhaps, read "...if they were planning to)."; - the sentence on page 16 on lines 44-45 should, perhaps, read "...any such events are reported to all principal investigators..."; - the sentence on page 18 on lines 44-45 should, perhaps, read "...where there are as few barriers as possible ..."; - in Table 3 on lines 17-18 there should be a space between "is" and "2)". - in Table 3 the abbreviation 'VHT' on lines 45-46 is neither used nor explained elsewhere in the manuscript.
--	---

REVIEWER	Carmen Logie University of Toronto, Canada
REVIEW RETURNED	20-Nov-2016

GENERAL COMMENTS	This is a well written, clear and methodologically strong protocol for a HIV self testing study (HIVST). The outcomes and design are well
---

described and justified. The authors do an excellent job of discussing endpoint measurement and steps to address social desirability bias. Overall this study has the potential to contribute to knowledge that can inform HIV testing interventions in Zambia and other regions. I provide some comments below:

Introduction

-It may be helpful for the reader if you could articulate the multiple and intersecting forms of stigma experienced by sex workers that influence their experience of HIV testing, including sex work stigma, HIV stigma, and larger contexts of gender inequity and violence.

-It may also be helpful to even briefly explain what you mean by stigma and discuss types (enacted, perceived stigma) across the multiple levels you mention in your discussion of HIV vulnerability. You mention stigma briefly, at the bottom of the first paragraph and again in the second paragraph as a barrier to testing. If you could define, and then highlight how sex workers accessing HIV testing experience intersecting stigma based on HIV as well as sex work that can be enacted at the institutional level by healthcare providers, but could also be realized at the community level. Sex workers may experience stigma and social exclusion based on being sex workers—but also stigma and social exclusion among sex workers if they test HIV positive, as well as the larger community. Describing how these larger social and structural contexts of intersecting HIV and sex work stigma may also influence the intrapersonal level (internalized stigma, uptake of HIV testing) is congruent with your multi-level framework.

-Theory of change:

-By further articulating what elements of stigma are relevant to understanding HIVST among your population in the introduction, you can articulate what kind/types of stigma you are referring to in your theory of change (“we theorized that the distribution of HIV self-test kits via peer educators would lead to improved status knowledge by reducing barriers to HIV testing such as stigma”). What specific stigma barriers are removed by HIVST? It seems as if experiences of healthcare provider stigma towards sex workers (enacted/perceived sex worker stigma at the institutional level) could be reduced; if this is what is hypothesized it would be helpful for the reader for this to be described. However, it is also possible that peer educators delivering HIV testing kits may not reduce HIV-related stigma at the community level (due to perceived HIV-related stigma); are the peer educators HIV-positive? What is HIV-related stigma like in the FSW communities in your context? In Caribbean and North American FSW communities there can be HIV-related stigma among FSW, and this is a deterrent to HIV testing. Known HIV-positive status can change the ability of FSW to work and acquire clients based on HIV-related stigma. Could this also be the case in your context, or is there less HIV-related stigma within sex work communities? Will the FSW be concerned about the peers finding out their HIV-positive status? How will this be mitigated?

Methods

-What dimensions of empowerment are you measuring? What is your hypothesis regarding empowerment (that empowerment scores will increase for the HIVST participants?)

-If you are including HIV-related stigma or sex work stigma as reasons for reduced uptake of HIV testing in your rationale, and exploring these issues in the qualitative component, it would be helpful to explain why you are not measuring either of these

	constructs in the survey(e.g. you may find outcomes vary by level of HIV stigma or sex work stigma). Discussion When you screen participants for IPV will they still be eligible to participate? If yes, how will you protect their safety when they take the HIVST? Overall this is an interesting, important and thoughtful study and I look forward to learning about the results.
--	--

VERSION 1 – AUTHOR RESPONSE

Reviewer: 1

Reviewer Name: Kaja-Triin Laisaar, MD, MPH, PhD

Institution and Country: Institute of Family Medicine and Public Health, University of Tartu, ESTONIA

Competing Interests: None declared

The study will provide important information on options to improve HIV testing frequency and HIV status knowledge in a high HIV risk group of female sex workers in Zambia. However, the protocol still needs some refinement.

Please see below major comments, following the manuscript 'Review Checklist' numeration, and a few minor comments.

Major comments:

2. Information in the abstract is not correct, as it is said that "...Participants (N=900) are eligible if they are women who have exchanged sex for money or goods in the previous 3 months,...", while elsewhere in the manuscript the period is limited to "last month" or "past month" (on page 9 and in Table 1, respectively).

Thank you for catching this error. The inclusion criterion is previous one month. We have corrected it (Page 2, Line 11):

"...exchanged sex for money or goods in the previous one month..."

4.1. Randomization information provided in the "Study design" paragraph (on page 7) and in Figure 1 (on page 32) is controversial -- it remains unclear whether there will be 5 or 6 FSW per peer educator.

While the target goal is 6 FSW participants per peer educator, we allow 5 per peer educator if a particular peer educator is having difficulty recruiting a total of 6 (so as she may proceed with the study in a timely fashion). We have clarified this in the Study Design section (Page 8, Line 139):

"...with a target enrollment of six FSW per peer educator (with five allowed if there are recruitment difficulties)."

In case the educators are also considered participants in this study, their recruitment, informed consent provision, etc. should be described in more detail.

Peer educators are not study participants in the trial. They are hired and paid as study staff, and no data is collected from them as part of the trial. We have clarified this in the Methods.

And even in case the educators are not considered participants, I would like to know more about how

they will be recruited -- to get an idea of a potential external validity issue [e.g., educators with organized (establishment-based) vs individual sex work experience recruiting FSW with similar background].

We agree this information will be useful. However, because we did not collect data from the peer educators, we do not have data on what type of venue(s) they work on. We have added additional detail about the recruitment of the peer educators to the "Recruitment" section (Page 10, Line 175):

"The peer educators are current or former FSW themselves who were recruited by study staff and who have undergone a two-day training in study procedures and HIV self-testing. Peer educators, who are not considered participants in this study, were recruited from current and former sex work organizations working in each of the study towns."

4.2. While describing the study procedures (starting on page 10) the reader would benefit from referral to Table 2 (in addition to already being referred to Table 1 and Figure 3), currently not mentioned anywhere in the text.

We agree with the reviewer and have added a reference to Table 2 where the reviewer suggests.

I would also suggest to use the study visit names provided in Table 2 in the "Study Time Point" column, as otherwise it would not be obvious (while reading the text) what/when is done. This also applies to Figure 1.

We agree that it would be useful to include the names in Table 2 in the text. We have changed these study visit names accordingly.

4.3. While reading the overview of the "Peer Educator Visits" (on page 10), I would have expected a few more words about the group-based visits -- how (first of all where) will these be organized.

Only the first peer educator visit is group based; the remainder are one-on-one visits with the peer educator. We have included additional details about the group-based visit (Page 12, Line 216):

"The initial group-based intervention is organized by the peer educator with assistance from the research assistant, and occurs in a private location of mutual convenience for the group members (such as study office, or near a work place)."

4.4. Somewhere in the "Study Procedures" paragraph I would also expect a few words about the 24/7 help-line -- will it be created for the study purposes only or is it locally available regardless of the study?

The hotline was created solely for the purpose of the study and is not available outside of the study. We have clarified this in the Study Procedures section (Page 12, Line 230):

"In particular, participants in all study arms also have access to an anonymous 24-hour hotline provided via the study. Participants will be instructed by the research assistant that they can call the 24-hour hotline at any time if they have difficulty with using the HIV self-test kit, need counseling related to HIV testing or gender-based violence, or need any other form of assistance over the duration of the study."

4.5. In the "Data collection" paragraph (on page 13), it should be described in a few words whether data in the follow-up assessments will also be "collected directly on a tablet in the field" (as in the baseline survey). This also applies to the in-depth qualitative interviews.

All quantitative assessments (baseline and follow-up) are collected via a tablet at a location that is convenient for the participant (in the field or in the study office). Qualitative assessments are collected on an audio recorder at the same location. We have added this to the “Data collection” section (Page 14, Line 276):

“Quantitative data are collected directly on a tablet.”

Page 16, Line 302:

“All qualitative data are collected via audio recorder.”

And (Page 15, Line 286):

“As with the baseline survey, all quantitative data will be collected via tablet, and qualitative data (Four Month Visit only) are collected via audio recorder.”

5. Please see my comment 4.1.

Please see response to comment 4.1.

7. How will data from qualitative interviews be analysed?

We have added additional detail about analysis for the qualitative data (Page 20, Line 389):

“Qualitative data will be analyzed using an iterative descriptive approach to characterize and describe the data at their natural level. After all interviews are transcribed and translated, a codebook will be finalized and transcripts will be coded in Dedoose, a cloud-based platform for mixed methods research.”

12. Please see my comment 4.1. regarding external validity.

Please see response to comment 4.1.

Also, as data management (as described in item 19 of the SPIRIT checklist) and confidentiality (as described in item 27 of the SPIRIT checklist) procedures are not described in detail, one might argue that not all the potential limitations of the study have been discussed in the manuscript.

We have added sections on Data Management (Page 21, Line 419)

“Data Management

Quantitative data are collected directly on a study tablet via CommCare (Dimagi, Cambridge, MA), a HIPAA-compliant, cloud-based platform for electronic data collection. Qualitative data are collected via audio recorder, which are transcribed and translated verbatim. Qualitative audio recordings are destroyed upon completion of transcription and translation.”

and Confidentiality (Page 20, Line 411).

“Confidentiality

To protect participants’ confidentiality, no identifying information will be collected during any quantitative or qualitative assessment and all collected data are de-identified. A unique study

identification number will be used to link data from an individual participant. Participant identity codes are kept completely separately from electronic data, in a locked cabinet in study offices. The code will be destroyed upon completion of the study.”

13. The SPIRIT Checklist seems to be filled out in a bit of hurry. While in general the numbers of pages provided might have shifted because of the title page automatically added to the manuscript submission package, for several items not all the applicable page numbers have been provided. Just two examples: item 5c (22-23), item 22 (11, 16, 20, Tables 2 and 3).

We have updated the SPIRIT Checklist in several ways, including ensuring all page numbers are provided.

Minor comments:

a. Regarding individual funding, reported by the authors on page 22, it is not clear whether the support was provided only in the past or will continue throughout the planned study;

Individual support will continue throughout the duration of the project for the majority of these sources of support. We have edited this section to read in the present tense, rather than past tense, to reflect this.

b. b. Figure 1 (on page 32) is missing a title;

We have included a title for Figure 1 on page 32 of the main text, under “Figure Legends”, rather than in the figure document itself.

c. The references should be reviewed, as:

- in case of more than three authors, in some cases all the authors have been listed, while in others 'et al.' is used (e.g., reference 2 vs reference 3);

We have updated the reference style to follow the guidelines for BMJ Open.

- in several cases the number of the journal issue is missing (e.g., reference 8);

We have corrected these references.

- in cases the reference is available online, a link might be provided (e.g., reference 11).

We have included a link.

d. There seem to be a few typos in the text:

- the sentence on page 16 on lines 24-25 should, perhaps, read "...if they were planning to).";

- the sentence on page 16 on lines 44-45 should, perhaps, read "...any such events are reported to all principal investigators...";

- the sentence on page 18 on lines 44-45 should, perhaps, read "...where there are as few barriers as possible ...";

- in Table 3 on lines 17-18 there should be a space between "is" and "2)".

- in Table 3 the abbreviation 'VHT' on lines 45-46 is neither used nor explained elsewhere in the manuscript.

Thank you for catching these. We have corrected them and proof-read the manuscript for additional

typos.

Reviewer: 2

Reviewer Name: Carmen Logie

Institution and Country: University of Toronto, Canada

Competing Interests: None declared

This is a well written, clear and methodologically strong protocol for a HIV self testing study (HIVST). The outcomes and design are well described and justified. The authors do an excellent job of discussing endpoint measurement and steps to address social desirability bias. Overall this study has the potential to contribute to knowledge that can inform HIV testing interventions in Zambia and other regions. I provide some comments below:

Introduction

-It may be helpful for the reader if you could articulate the multiple and intersecting forms of stigma experienced by sex workers that influence their experience of HIV testing, including sex work stigma, HIV stigma, and larger contexts of gender inequity and violence.

We agree that it would be useful to discuss the multiple and intersecting forms of stigma that may be influencing HIV testing. We have added to the Introduction (Page 6, Line 79):

“For FSW, stigma can be due to engagement in sex work itself, or from HIV stigma, particularly in contexts of high HIV burden, and can come from family or other community members, partners, and healthcare providers. These experiences of stigma may be exacerbated by unequal power dynamics, poverty and financial reliance on sex work, and experiences of violence.”

-It may also be helpful to even briefly explain what you mean by stigma and discuss types (enacted, perceived stigma) across the multiple levels you mention in your discussion of HIV vulnerability. You mention stigma briefly, at the bottom of the first paragraph and again in the second paragraph as a barrier to testing. If you could define, and then highlight how sex workers accessing HIV testing experience intersecting stigma based on HIV as well as sex work that can be enacted at the institutional level by healthcare providers, but could also be realized at the community level. Sex workers may experience stigma and social exclusion based on being sex workers—but also stigma and social exclusion among sex workers if they test HIV positive, as well as the larger community. Describing how these larger social and structural contexts of intersecting HIV and sex work stigma may also influence the intrapersonal level (internalized stigma, uptake of HIV testing) is congruent with your multi-level framework.

We agree. We have added a paragraph to the Introduction that specifically discusses the importance of stigma for FSW when it comes to barriers to HIV testing (Page 6, Line 74):

“Stigma and discrimination can affect access to HIV testing among FSW. 7-10 FSW may be affected by multiple, intersecting forms of stigma. Stigma is broadly defined as negative attitudes, relative powerlessness, and loss of status related to a particular characteristic. Stigma can be experienced as enacted (explicit actions such as derogatory language, active discrimination, or assault), perceived (the expectation of enacted stigma), or self-stigma (internalization of stigma). For FSW, stigma can be due to engagement in sex work itself, or from HIV stigma, particularly in contexts of high HIV burden, and can come from family or other community members, partners, and healthcare providers. These experiences of stigma may be exacerbated by unequal power dynamics, poverty and financial reliance on sex work, and experiences of violence. Taken together, the multidimensionality of experiences of stigma and discrimination may greatly influence uptake of HIV testing. Interventions that mitigate stigma as a barrier to HIV testing may be especially powerful for FSW.”

-Theory of change:

-By further articulating what elements of stigma are relevant to understanding HIVST among your population in the introduction, you can articulate what kind/types of stigma you are referring to in your theory of change (“we theorized that the distribution of HIV self-test kits via peer educators would lead to improved status knowledge by reducing barriers to HIV testing such as stigma”). What specific stigma barriers are removed by HIVST? It seems as if experiences of healthcare provider stigma towards sex workers (enacted/perceived sex worker stigma at the institutional level) could be reduced; if this is what is hypothesized it would be helpful for the reader for this to be described.

We agree that it is primarily healthcare provider stigma that provision of HIVST may mitigate. We have added to this section (Page 8, Line 128):

“Enacted or perceived sex work stigma from healthcare providers and from the community may be addressed by HIVST, by allowing individuals to test for HIV in private without fear of being seen in the clinic and without fear of judgment from providers.”

However, it is also possible that peer educators delivering HIV testing kits may not reduce HIV-related stigma at the community level (due to perceived HIV-related stigma); are the peer educators HIV-positive? What is HIV-related stigma like in the FSW communities in your context? In Caribbean and North American FSW communities there can be HIV-related stigma among FSW, and this is a deterrent to HIV testing. Known HIV-positive status can change the ability of FSW to work and acquire clients based on HIV-related stigma. Could this also be the case in your context, or is there less HIV-related stigma within sex work communities?

We did not have HIV status criteria for the peer educators, nor did we ask them their HIV status as part of working in this study. However, it is quite likely that some of the peer educators are living with HIV. There is some degree of HIV-related stigma among FSW in our context, and they do report “fear” as being a barrier to HIV testing. We have added to this section (Page 8, Line 133):

“However, it is also possible that a community-based intervention such as HIVST could be unsuccessful if individuals are concerned about others discovering their HIV status.”

Will the FSW be concerned about the peers finding out their HIV-positive status? How will this be mitigated?

It is possible that FSW will be concerned about HIV positive status. We have attempted to design the study such that peer educators will not necessarily learn the status of their group members unless the group member wants to disclose to the peer educator. Participants are interviewed separately from the peer educator, and are told that they can contact the peer educator and/or the hotline for help, but the peer educator intervention does not include discussion of test results nor do participants test with the peer educator. We have added to the Methods (Page 12, Line 227):

“Peer educators do not specifically discuss HIV status with any participant unless the participant seeks out the peer educator for advice. Participants are given alternatives, such as the study hotline, should they need help with HIVST or another element of the study but do not want to disclose their HIV status to the peer educator.”

Methods

-What dimensions of empowerment are you measuring? What is your hypothesis regarding empowerment (that empowerment scores will increase for the HIVST participants?)

We are measuring empowerment both quantitatively and qualitatively. Quantitatively, we are measuring sex worker agency and power within. Qualitatively, we are exploring perceived control over lives, ability to change situations, and community empowerment. We hypothesize that access to HIVST will increase empowerment scores by improving agency for HIV testing. We have added to the Methods (Page 18, Line 363):

“Quantitatively, empowerment is measured with a 5-item scale addressing sex worker agency and power within. Qualitatively, empowerment is measured as ability to get things the participant needs, to change her situation, and control over elements such as condom use. We hypothesize that access to HIVST will lead to an improved sense of empowerment.”

-If you are including HIV-related stigma or sex work stigma as reasons for reduced uptake of HIV testing in your rationale, and exploring these issues in the qualitative component, it would be helpful to explain why you are not measuring either of these constructs in the survey(e.g. you may find outcomes vary by level of HIV stigma or sex work stigma).

We apologize for the oversight, we are actually measuring both sex work-related and HIV-related stigma in our quantitative surveys using two separate stigma scales. We have clarified this in the Methods (Page 18, Line 359):

“Quantitatively, sex work stigma is measured via a 6-item scale that covers enacted, perceived, and self-stigma, and HIV stigma is measured by a 9-item scale measuring attitudes towards individuals living with HIV.”

Discussion

When you screen participants for IPV will they still be eligible to participate? If yes, how will you protect their safety when they take the HIVST?

Participants who have experienced IPV will still be eligible to participate. IPV in this population is very high (~60% in the baseline survey), and comes from multiple perpetrators, including clients and non-commercial intimate partners. Given the extraordinarily high prevalence of IPV in this population, we felt that it was important to include women who had experienced IPV. This intervention specifically does not include a partner-testing component to mitigate IPV (although such research questions are of interest, we specifically chose not to include partner testing for this reason). During the informed consent process, potential risks including physical or psychological harm are covered. Participants have access to the peer educators, the hotline, and study staff and are given referrals to counseling in the event IPV does occur. They are additionally counseled to use the test in a place that is private and where they feel safe, although ultimately where they choose to use it or who they choose to talk to about it is up to them.

We have added additional language to the Discussion (Page 24, Line 483):

“Intimate partner violence is very common in the FSW community, and those who have experienced intimate partner violence will be eligible to participate, although the risks of participating are explained fully. Furthermore, during follow-up peer education visits, peer educators will screen for intimate partner violence, and they will also be available between visits should any issues come up. Finally, a 24-hour telephone line will be available to all participants for any issues, including intimate partner violence, and referral will be made to counseling if desired.”

Overall this is an interesting, important and thoughtful study and I look forward to learning about the

results.

Thank you for your comments.

VERSION 2 – REVIEW

REVIEWER	Kaja-Triin Laisaar, MD, MPH, PhD Institute of Family Medicine and Public Health, University of Tartu, ESTONIA
REVIEW RETURNED	23-Jan-2017

GENERAL COMMENTS	This protocol for a cluster randomized trial on HIV self-testing among female sex workers in Zambia is methodologically strong, providing sufficient detail of the planned study. Just a tiny remark: On page 6 (line 71 in the pdf document), following the sentence 'Stigma and discrimination can affect access to HIV testing among FSW.', the references should be presented in square brackets.
---

REVIEWER	Carmen Logie University of Toronto, Canada
REVIEW RETURNED	08-Feb-2017

GENERAL COMMENTS	The authors did an excellent job revising the manuscript. The only comment I have is that there is not a comprehensive definition of empowerment provided (just how it could be measured quantitatively and qualitatively), and although there is a hypothesis about empowerment ('We hypothesize that access to HIVST will lead to an improved sense of empowerment') it does not map onto any of the specific trial aims. There is not a specific hypothesis about HIV or sex work stigma either, and it also is not mapped onto any aims or objectives. Perhaps adding a secondary aim of understanding stigma and empowerment and its association with HIVST, would help the reader understand the inclusion of these very important outcomes. I look forward to hearing about the results of this innovative work.
--

VERSION 2 – AUTHOR RESPONSE

Reviewer: 1

Reviewer Name: Kaja-Triin Laisaar, MD, MPH, PhD

Institution and Country: Institute of Family Medicine and Public Health, University of Tartu, ESTONIA

Competing Interests: None declared

This protocol for a cluster randomized trial on HIV self-testing among female sex workers in Zambia is methodologically strong, providing sufficient detail of the planned study.

Just a tiny remark: On page 6 (line 71 in the pdf document), following the sentence 'Stigma and discrimination can affect access to HIV testing among FSW.', the references should be presented in square brackets.

We thank the reviewer for their comments. We have edited this so that the references are in square

brackets.

Reviewer: 2

Reviewer Name: Carmen Logie

Institution and Country: University of Toronto, Canada

Competing Interests: None declared

The authors did an excellent job revising the manuscript. The only comment I have is that there is not a comprehensive definition of empowerment provided (just how it could be measured quantitatively and qualitatively), and although there is a hypothesis about empowerment ('We hypothesize that access to HIVST will lead to an improved sense of empowerment') it does not map onto any of the specific trial aims.

We thank the reviewer for their comments. We have added a definition of empowerment (Page 17, Line 357):

“Empowerment is defined as the process by which disenfranchised individuals acquire the ability to make choices.[24]”

There is not a specific hypothesis about HIV or sex work stigma either, and it also is not mapped onto any aims or objectives. Perhaps adding a secondary aim of understanding stigma and empowerment and its association with HIVST, would help the reader understand the inclusion of these very important outcomes. I look forward to hearing about the results of this innovative work.

We agree with the reviewer that the stigma outcomes are important. Stigma was not a prespecified outcome of the trial, so we did not include it in our prespecified specific aims. We primarily were interested in stigma as a barrier to existing HIV testing, and were interested in effect modification of HIVST uptake by baseline stigma levels. We have included in the Methods (Page 17, Line 355):

“We hypothesize that at baseline individuals who report a greater amount of stigma will have reduced HIV testing, and that these individuals will have greater uptake of HIVST.”